# Leptin, Ghrelin, Irisin, Asprosin and Subfatin Changes in Obese Women: Effect of Exercise and Different Nutrition Types

**DOI:** 10.3390/medicina60071118

**Published:** 2024-07-10

**Authors:** Elif Bengin, Abdurrahman Kırtepe, Vedat Çınar, Taner Akbulut, Luca Russo, İsa Aydemir, Polat Yücedal, Süleyman Aydın, Gian Mario Migliaccio

**Affiliations:** 1Institute of Health Sciences, Faculty Sport Science, Firat University, Elazig 23200, Turkey; elifbengin@hotmail.com; 2Department of Physical Education and Sport, Faculty Sport Science, Firat University, Elazig 23200, Turkey; akirtepe@firat.edu.tr (A.K.); cinarvedat@hotmail.com (V.Ç.); 3Department of Coaching Education, Faculty Sport Science, Firat University, Elazig 23200, Turkey; takbulut@firat.edu.tr; 4eCampus University, 22060 Novedrate, Italy; 5Department of Physical Education and Sport, Faculty of Education, Hakkari University, Hakkari 30100, Turkey; aydemirisa23@gmail.com; 6Department of Coaching Education, Faculty Sport Science, Munzur University, Tunceli 62100, Turkey; yucedalpolat@gmail.com; 7Department of Biochemistry, Faculty of Medicine, Firat University, Elazig 23200, Turkey; saydin1@hotmail.com; 8Department of Human Sciences and Promotion of the Quality of Life, San Raffaele Rome Open University, 00166 Rome, Italy; gianmario.migliaccio@uniroma5.it

**Keywords:** diet, exercise, nutrition, asprosin, ırisin

## Abstract

*Background and Objectives:* In this study, the effects of a six-week training program and various diets on subfatin, asprosin, irisin, leptin, ghrelin and the lipid profile were investigated in overweight women. *Materials and Methods:* A total of 78 women voluntarily participated in the study. Groups: The study was divided into eight groups: Healthy Control, Obese Control, Obese + Vegetarian, Obese + Ketogenic, Obese + Intermittent Fasting, Obese + Exercise + Vegetarian, Obese + Exercise + Ketogenic and Obese + Exercise + Intermittent Fasting. While there was no intervention in the healthy and obese control groups, the other groups followed predetermined exercise and diet programs for 6 weeks. Blood samples were taken from the participants in the research group twice (before and after the interventions). An autoanalyzer was used to determine the lipid profile in the blood samples taken, and the ELISA method was used to analyze other parameters. *Results:* Overall, a significant difference was found in the values of weight, BMI, subfatin, ghrelin, leptin, cholesterol, triglyceride, HDL and LDL as a result of the exercise and diet interventions (*p* < 0.05). There was no significant difference in asprosin and irisin values (*p* > 0.05). *Conclusions:* In conclusion, regular exercise and dietary interventions in obese women can regulate lipid profile, ghrelin, leptin and asprosin levels, and increasing irisin with exercise can activate lipid metabolism and support positive changes in lean mass.

## 1. Introduction

Obesity is defined by the World Health Organization (WHO) as excessive fat accumulation in the body to the extent that it impairs health. Although other factors may be involved, the main cause of obesity is the imbalance between the amount of calories taken in and the amount of calories spent. Obesity is accepted as an individual’s body mass index (BMI) being higher than 30 [1]. In addition, anthropometric measurements such as skinfold thickness, waist circumference and waist–hip ratio have become increasingly used to assess an individual’s risk of obesity-related conditions such as T2DM and cardiovascular diseases [2]. Age, sex, eating habits, sociocultural factors, physical inactivity and genetic factors all play an important role in the formation of obesity. Determining the factors affecting the frequency of obesity is important for solving possible health problems and taking necessary precautions [3,4,5,6]. Although genetic factors are responsible for 30–70% of obesity, high-calorie diet consumption, changing lifestyle and decreased physical activity levels lead to a decrease in energy consumption and an increase in obesity, especially in developed societies [4,7]. Energy balance is regulated by various biomolecules. A group of hormones (such as leptin and ghrelin) are involved in maintaining a balanced appetite, energy and body weight. These hormones control hunger and satiety via neuronal pathways and determine the frequency and amount of eating [8]. For this reason, the tissues in which these hormones are released are also important. Adipose tissue is now considered an important part of energy metabolism. Adipokines are secreted in white adipose tissue [9]. Adipokines play a role in many physiological processes of the organism, such as appetite, energy balance, insulin and glucose metabolism, lipid metabolism and blood pressure regulation. One of these, asprosin, produced by adipocytes in white adipose tissue, provides glucose production and release [10,11]. At the same time, asprosin can control appetite in the hypothalamus region of the brain [12]. However, irisin, defined as myokine, is a hormone thought to play an active role in the prevention and treatment of obesity and metabolic syndrome. The most important factor in stimulating the irisin is exercise. However, the type, intensity and duration of the exercise can change the level of the irisin at this point [13,14]. The hormones leptin, ghrelin and insulin are also effective in the long-term regulation of appetite. Leptin released from adipose tissue and insulin released from B-cells of the pancreas are hormones whose synthesis increases in proportion to the increase in body fat mass and suppress appetite in the hypothalamus. Ghrelin hormone released from the stomach activates orexigenic neurons, increasing appetite and reducing energy expenditure [15]. Blood lipids, where the effects of exercise and nutrition have been extensively studied, include cholesterol, HDL (high-density lipoprotein), LDL (low-density lipoprotein) and triglycerides [16]. These are important elements for assessing cardiovascular health. In addition, Meteorin-Like (subfatin) is an adipomyokine secreted from muscle tissues during exercise. One of its most important functions is to improve blood glucose tolerance and increase insulin sensitivity [17]. These topics provide important information on the course of obesity. The effectiveness of different nutritional models applied together with exercise on the mentioned parameters is a matter of curiosity. For this reason, it was hypothesized in the study that the combined application of exercise and nutrition models on obese women may affect the biochemical elements associated with obesity. All information in the present study was aimed to determine the effects of an exercise program supported by different diet types on body weight, BMI and biochemical values (leptin, ghrelin, asprosin, sunfatin, irisin, lipid profile).

## 2. Materials and Methods

The research was conducted in accordance with the pre-test–post-test control group research model (Figure 1). The research was voluntary, and obese women who did not have any disabilities for the exercise practices were included in the study. Before starting the research, ethics committee approval was received from the Fırat University Non-invasive Research Ethics Committee with session date 7 June 2022 and session number 2022/08-02. Additionally, the study was conducted in accordance with the Declaration of Helsinki. Exclusion criteria from the study, such as participants’ desire to leave the study, failure to comply with exercise programs and illness or injury during this process, were determined. In addition, failure to comply with the specified diet lists was accepted as a criterion for exclusion from the study. In this context, a total of 78 obese women with an average age of 33.97 ± 9.77 and an average height of 159.15 ± 3.91 participated in the study voluntarily. Participants were divided into 8 groups (Table 1).

### 2.1. Applied Exercise Protocol

In our research, the aerobic exercise program specified in Table 2 was applied to the participants in the exercise groups 3 days a week and 60 min a day for 6 weeks. Since our study group consists of overweight individuals and has risk factors, they were examined by a doctor, and volunteer consent forms were obtained. Exercise planning was created to keep the target heart rate at 50–60%. At the end of the study, a 5–10 min cooling and stretching exercise was performed.

### 2.2. Diet Protocols 

The diet list of each group was changed weekly during the 6-week period. The list of groups was prepared by taking into account people’s age, height, socioeconomic status, lifestyle and criteria for compliance with the list. Daily meal and nutrition tracking was performed online. 

#### 2.2.1. Vegetarian Nutrition Protocol

Do not consume red meat and consume limited amounts of chicken and fish. It was implemented with a program based on plant-based foods such as grains, legumes, vegetables, fruits and oilseeds. In a study investigating compliance with a low-fat vegetarian diet, it was stated that although compliance with the diet varies in some studies, it is generally easy to comply with the diet [18]. The semi-vegetarian diet model was preferred because it is the closest diet to routine life. Having a high fiber source in the diet provides an advantage in adjusting glycemic control in people with diabetes and losing weight in obese people [19].

#### 2.2.2. Ketogenic Nutrition Protocol

Ketogenic diets aim to achieve ketosis as soon as possible [20,21]. The standard ketogenic diet type is applied, and the diet generally contains 75 percent fat, 20 percent protein and only 5 percent carbohydrates. In the ketogenic diet, the consumption of foods high in suitable-quality fat and protein is allowed, while the consumption of foods containing carbohydrates is limited to the specified extent. Consumption of grain products such as wheat, barley, rye, oats, corn and rice, products containing simple sugars, fruits and fruit juices, root vegetables such as potatoes, and legumes such as chickpeas, beans and lentils are limited in terms of their carbohydrate content.

#### 2.2.3. Intermittent Fasting Nutrition Protocol

In our research, the 16:8 module, that is, 16 h without eating and then 8 h again with food, was preferred. A diet consisting of two main meals, breakfast and dinner, was implemented within 24 h, and sleeping hours were included in the 16 h fast. Solid foods were not consumed for 8 h between morning and evening meals. Only non-caloric liquids such as water, plain soda, unsweetened coffee or tea are allowed. This nutritional model has positive effects on insulin, glucose levels and lipid levels [22].

### 2.3. Determination of Body Weight and BMI

AVIS 333 Bioelectric Impedance device was used to determine the body weights and BMI values of the participants.

### 2.4. Taking and Analysis of Blood Samples

To be used in the analyses, 5 mL of blood was taken from the participants in the morning after a 12 h overnight fast, at the beginning and at the end of the study, and was centrifuged at 4000 rpm for 5 min and stored at −80 degrees until the day of the study. Examples SunRed brand (Sunred Biological Technology Co., Ltd., Shanghai, China), subfatin (catalog no: 201-12-9252), asprosin, irisin (catalog no: 201-12-5328), leptin (catalog no: 201-12-1560), ghrelin (catalog no: 201-12-5583) were studied by ELISA method using kits. A Bio-tek ELX50 (BioTek Instruments, Winooski, VT, USA) automatic washer was used for washing processes. For absorbance readings, a ChroMate Microplate Reader P4300 (Awareness Technology Instruments, Palm City, FL, USA) ELISA reader was used at 450 nm. The lipid profile parameters (Cholesterol, HDL-Cholesterol, LDL-Cholesterol and Triglyceride) levels included in the research were determined using the CELL-DYN-3500 R brand autoanalyzer.

### 2.5. Statistical Analysis 

SPSS 22.0 package program was used to analyze the data obtained from the research. Accordingly, arithmetic mean, standard deviation and frequency distribution techniques were used as descriptive statistics in the analysis of the data. Wilcoxon test was used to determine intra-group differences before and after the nutrition–exercise protocol was applied, and analysis of variance (Kruskal–Wallis test) was performed to reveal inter-group differences, and statistical significance was accepted as *p* < 0.05. In addition, effect sizes (ES) obtained from the research results were calculated according to Cohen’s formula. Finally, the effect size obtained is as follows: ≤0.4 was considered a small effect, 0.41–0.70 was considered a medium effect and ≥0.70 was considered a large effect [23].

## 3. Results

When Figure 2 was examined, it was determined that the pre-test and post-test values of body weight and BMI variables differed in all groups except the OC group (*p* < 0.05). Considering the ES values, it is seen that the biggest change for both variables occurred in the O + IF + E and O + K + E groups, respectively.

According to Figure 3, while there is no significant difference between the cholesterol and HDL-Cholesterol pre-test–post-test values in the HC, OC and O + V groups (*p* > 0.05), there are differences in all other groups (*p* < 0.05). It seems that the group with the largest ES on both variables is O + IF + E (Cholesterol: 2.07, HDL-Cholesterol: 0.19). In Trigyliceride and LDL-Cholesterol values, it was determined that there was a significant difference between the pre- and post-test values of all groups except the HC and OC groups (*p* < 0.05). It seems that the group with the largest ES on both variables is O + K + E (Triglyceride: 3.82, LDL-Cholesterol: 0.74).

It was determined that there was no significant difference in the pre-test–post-test levels of leptin and ghrelin in the HC, OC, O + V and O + K groups. In addition, ghrelin level did not change in the O + K + E group (*p* > 0.05). It was observed that there were statistically significant differences between the pre-test and post-test values in all other groups (*p* < 0.05). It is understood that ES values correspond to a small effect in all groups (Figure 4).

Although the level of the irisin variable increased with exercise, this difference was not statistically significant (*p* > 0.05). It was determined that the asprosin variant differentiated in the O + V + E group, and subfatin differentiated in the O + K + E and O + IF + E groups (*p* < 0.05). It is understood that the ES values in these changes correspond to a small effect (Figure 5).

## 4. Discussion

The research was conducted to determine the effects of different nutrition types and exercise interventions on body weight, BMI, leptin, ghrelin, asprosin, subfatin, irisin and lipid profile in obese women. Research findings show that exercise and nutrition protocols create positive effects and changes in the experimental groups compared to the control groups. As it is known, nutrition and physical activity are the cornerstones of a healthy lifestyle. Current research results also reveal the importance of this situation. In addition to reducing body fat levels and protecting against obesity-related chronic diseases, regular exercise and sports are also known to have therapeutic roles. A study conducted in this context emphasized that individuals who exercise and eat healthy have a decrease in body weight, body mass index and body fat [24,25]. In the current study, it was determined that BMI and body weight decreased with exercise and diet. It has been observed that diet alone reduces BMI, but its effect is higher when applied together with exercise. Differences in body weight changes are due to the effects of complex feedback mechanisms such as genetic factors, differences in neurohormonal mechanisms, metabolic efficiency and tissue capacity, non-exercise-induced thermogenesis, the thermogenic effect of nutrients and intestinal microbiome on the adaptation of “energy restriction” [26]. Therefore, it would not be correct to attribute the body weight changes of both vegetarian individuals and other diet types only to the change in their diet. It is thought that the change in cholesterol, HDL, LDL and Triglyceride levels is due to the positive effects of exercise and a low-fat diet. A meta-analysis in this field compared the effects of 14 different diets on LDL and HDL levels. Despite significant weight loss, the effect of these diets on LDL and HDL levels is low [27]. The effect of exercise on serum LDL varies, with some studies showing a 4–7% decrease and some even increase [28,29]. The decrease in LDL levels typically occurs in conjunction with weight loss. A significant amount of exercise (700–2000 kcal of exercise per week) is required to significantly increase HDL levels [28,30]. In the current study, it can be said that HDL increased more, especially in the exercise group. A meta-analysis of 11 studies on the ketogenic diet found that the low-carb diet group experienced significant weight loss compared to the low-fat diet group. People on a very low-carb ketogenic diet (VLCKD) experienced decreases in body weight, triglycerides, and diastolic blood pressure, as well as increases in HDL and LDL. Moreover, VLCKD resulted in more significant long-term weight loss compared to the low-fat diet, suggesting that it is a potential alternative option for obesity management [31]. Body weight loss may vary even when factors such as diet compliance, physical activity, sex, age and specific medications, whose effectiveness is well known, are controlled [32].

As it is known, excess body fat causes more leptin secretion. It has been assumed that the suppression of ghrelin secretion is among the satiety-inducing effects of leptin [33]. High-intensity exercise suppresses appetite in adults, and this is thought to be related to hormones that regulate appetite. The increase in IL-6 caused by high-intensity exercise in both normal weight and overweight/obese boys has been associated with a decrease in appetite, and it has been stated that the decrease in active ghrelin and/or the increase in endogenous cortisol may be effective, even to a small extent, in suppressing appetite [34]. A study examined changes in serum ghrelin and leptin levels after 12 weeks of aerobic training and gonadotropin-releasing hormone agonist (GnRH) treatment in girls in early puberty. It was shown that aerobic training increased ghrelin and decreased leptin and the ratio of leptin to ghrelin [35]. In a study in which 62 obese and 48 healthy individuals participated, appropriate diet (1000–1500 kcal/day) and exercise (at least 5000 steps/day) programs were applied according to age, sex and BMI. The ghrelin values of the participants decreased significantly after 12 weeks with diet and exercise [36]. In the current study, a decrease in leptin value was observed in all groups, while a difference in ghrelin value was observed in every group except the Ketogenic and Exercise groups.

Blüher et al. stated that after a one-year diet + exercise program, there was a 12% increase in irisin levels of obese young people [37]. It has also been reported that chronic and acute exercise training leads to an increase in circulating irisin levels (plasma/serum) in healthy individuals, with a greater increase observed in acute aerobic and chronic resistance protocols [38,39]. Changes in irisin level with changing nutritional patterns are not clear. For this reason, debates regarding the metabolic effects of irisin and its role in obesity still continue [40]. A meta-analysis in this field revealed a significant increase in irisin levels following continuous endurance training [41].

Subfatin plays a regulatory role in energy expenditure and browning of adipose tissue [42]. Regular exercise can lead to an increase in fat tissue subfatin even in a chronically obese state. It shows that exercise-induced muscle subfatin level is effective in reducing fat accumulation by increasing subfatin in adipose tissue, which may be a therapeutic target for chronic obesity [43]. A study evaluating the therapeutic effect of treadmill exercise showed that exercise induces upregulation of subfatin in serum and synovial fluid [44]. Another study supporting this shows that subfatin can be controlled with exercise intervention [45]. In another study conducted on obese individuals, the correlations of subfatin level with anthropometric parameters, HOMA index and biochemical measurements were evaluated, and subfatin serum levels were found to be lower in obese patients than in healthy controls. It has been determined that the amount of circulating subfatin hormone may be a new biological marker of obesity and insulin resistance [46]. Adipokines are bioactive substances secreted from adipose tissue and have various functions in appetite, energy, lipid and carbohydrate metabolism, blood pressure regulation and inflammation. A recent study revealed the role of asprosin in alleviating the inflammatory response of macrophages and inducing the atheroprotection effect [47]. A hospital-based observational study consisting of 170 people showed a significant increase in serum asprosin levels in newly diagnosed T2DM adult patients compared to individuals with normal glucose tolerance [48]. In another study, BMI increased in both exercising and sedentary participants. It shows that the serum asprosin level increases with increasing BMI, and the serum asprosin level of obese participants was significantly higher than that of those with normal BMI. This suggests that exercise training cannot prevent a significant increase in serum [49]. In the current study, it was seen that there may be changes in the intermittent fasting diet and in the ketogenic diet and exercise group. This shows that the form of exercise and nutrition may be closely related.

### Limitations

As with every research, this research has some limitations. Most of the individuals included in the analysis were limited by short follow-up periods (6 weeks) and relatively small sample sizes. This is partly because weight loss strategies make it relatively difficult for participants to stick to a designated diet program for long periods of time, as well as because there is a lack of follow-up to assess continued beneficial effects after the dieting process is over. In addition, the fact that the study was conducted only with female participants and the absence of a group that only exercised can be considered as limitations of the study.

## 5. Conclusions and Recommendations

As a result, it has been observed that different nutrition types and exercises applied to obese women are effective on many of the determined parameters. These effects were seen clearly in subfatin, ghrelin, leptin, glucose, solubility, triglyceride, HDL and LDL values. Considering the relationship of these parameters with obesity and human health, the results obtained reveal the importance of diet and exercise combinations. This shows that the combination of nutrition and exercise is important in the prevention and treatment of obesity. Based on this, what needs to be done for a healthy life is to support body weight control with a diet pattern and exercise practice that is applicable to everyone. Additionally, as with any other type of diet, it is more important to focus on body composition for weight loss goals. However, it is understood that longer-term studies are needed for stronger results.

## Figures and Tables

**Figure 1 medicina-60-01118-f001:**
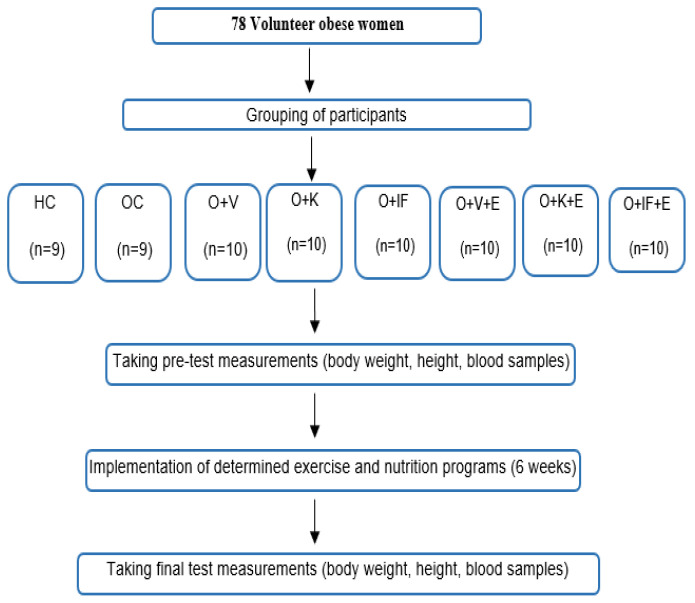
Research design.

**Figure 2 medicina-60-01118-f002:**
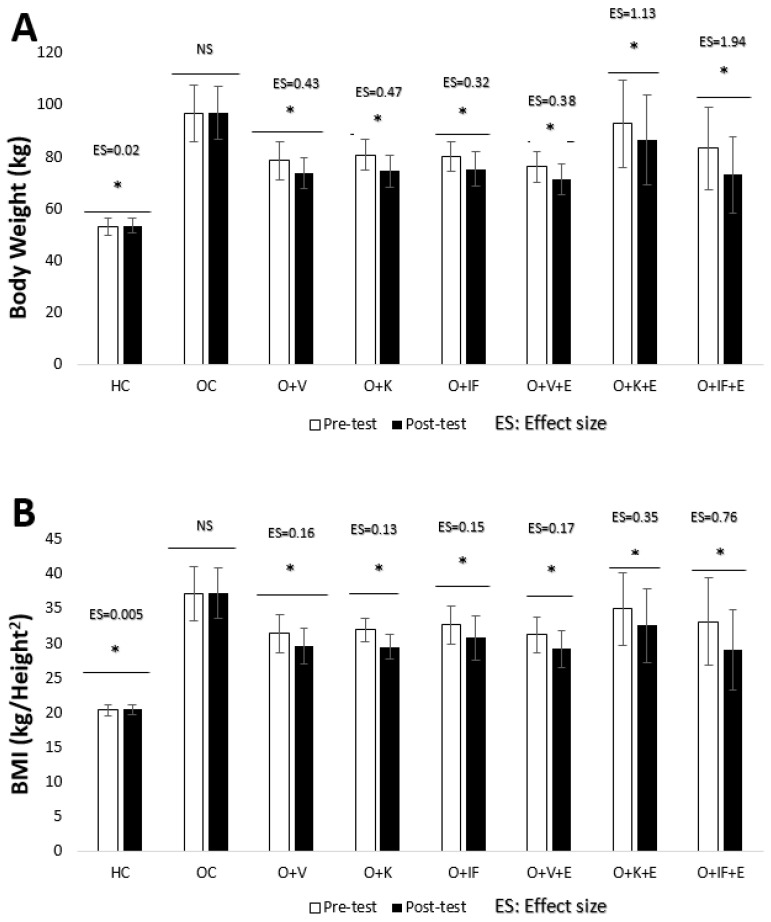
Body weight (**A**) and BMI (**B**) levels. Healthy Control (HC), Obese Control (OC), Obese + Vegetarian (O + V), Obese + Ketogenic (O + K), Obese + Intermittent Fasting (O + IF), Obese + Vegetarian + Exercise (O + V + E), Obese + Ketogenic + Exercise (O + K + E) and Obese + Intermittent Fasting+ Exercise (O + IF + E). * *p* < 0.05, NS; non-significant and ES; Effect Size.

**Figure 3 medicina-60-01118-f003:**
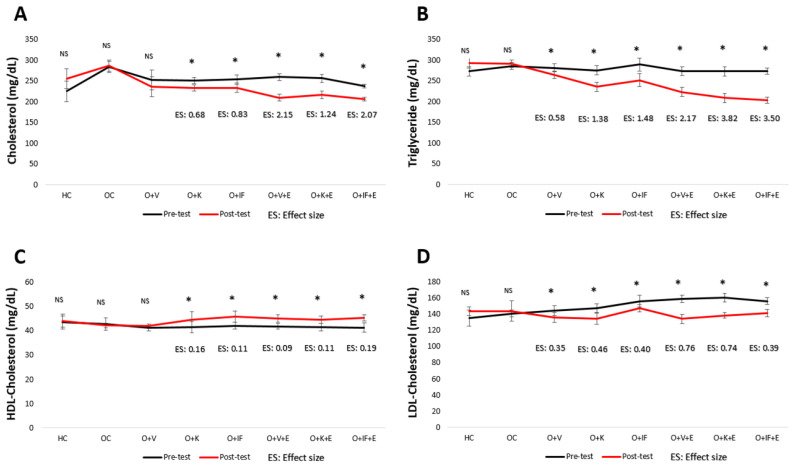
Cholesterol (**A**), Triygliceride (**B**), HDL-Cholesterol (**C**) and LDL-Cholesterol (**D**) levels. Healthy Control (HC), Obese Control (OC), Obese + Vegetarian (O + V), Obese + Ketogenic (O + K), Obese + Intermittent Fasting (O + IF), Obese + Vegetarian + Exercise (O + V + E), Obese + Ketogenic + Exercise (O + K + E) and Obese + Intermittent Fasting+ Exercise (O + IF + E). * *p* < 0.05, NS; non-significant and ES; Effect Size.

**Figure 4 medicina-60-01118-f004:**
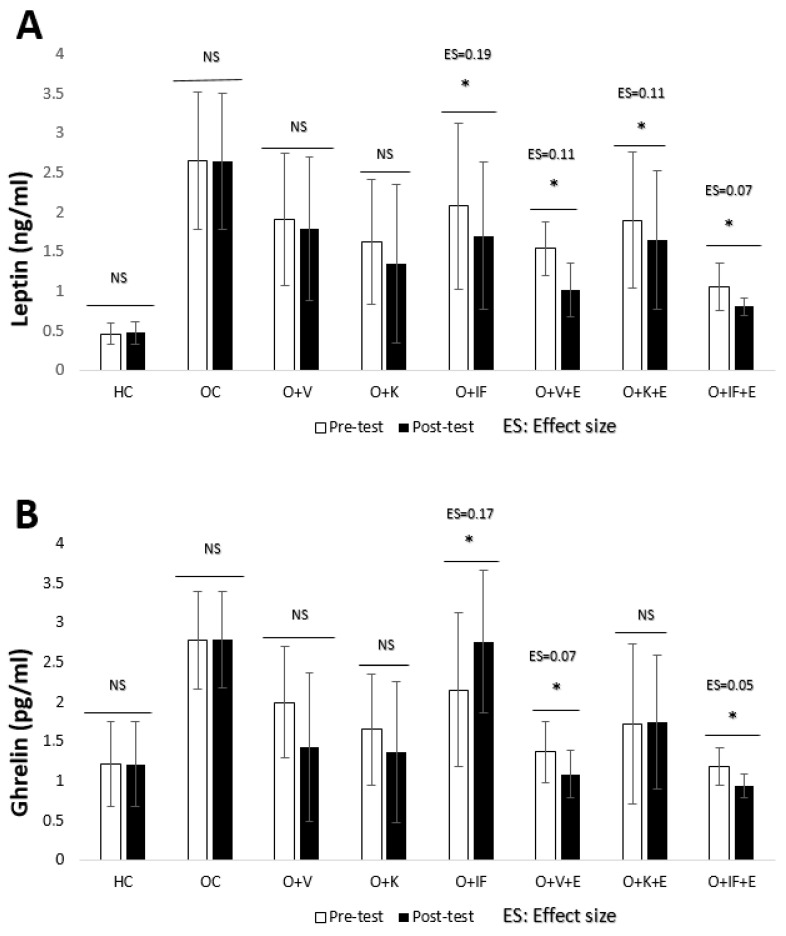
Leptin (**A**) and ghrelin (**B**) levels. Healthy Control (HC), Obese Control (OC), Obese + Vegetarian (O + V), Obese + Ketogenic (O + K), Obese + Intermittent Fasting (O + IF), Obese + Vegetarian + Exercise (O + V + E), Obese + Ketogenic + Exercise (O + K + E) and Obese + Intermittent Fasting+ Exercise (O + IF + E). *; *p* < 0.05, NS; non-significant and ES; Effect Size.

**Figure 5 medicina-60-01118-f005:**
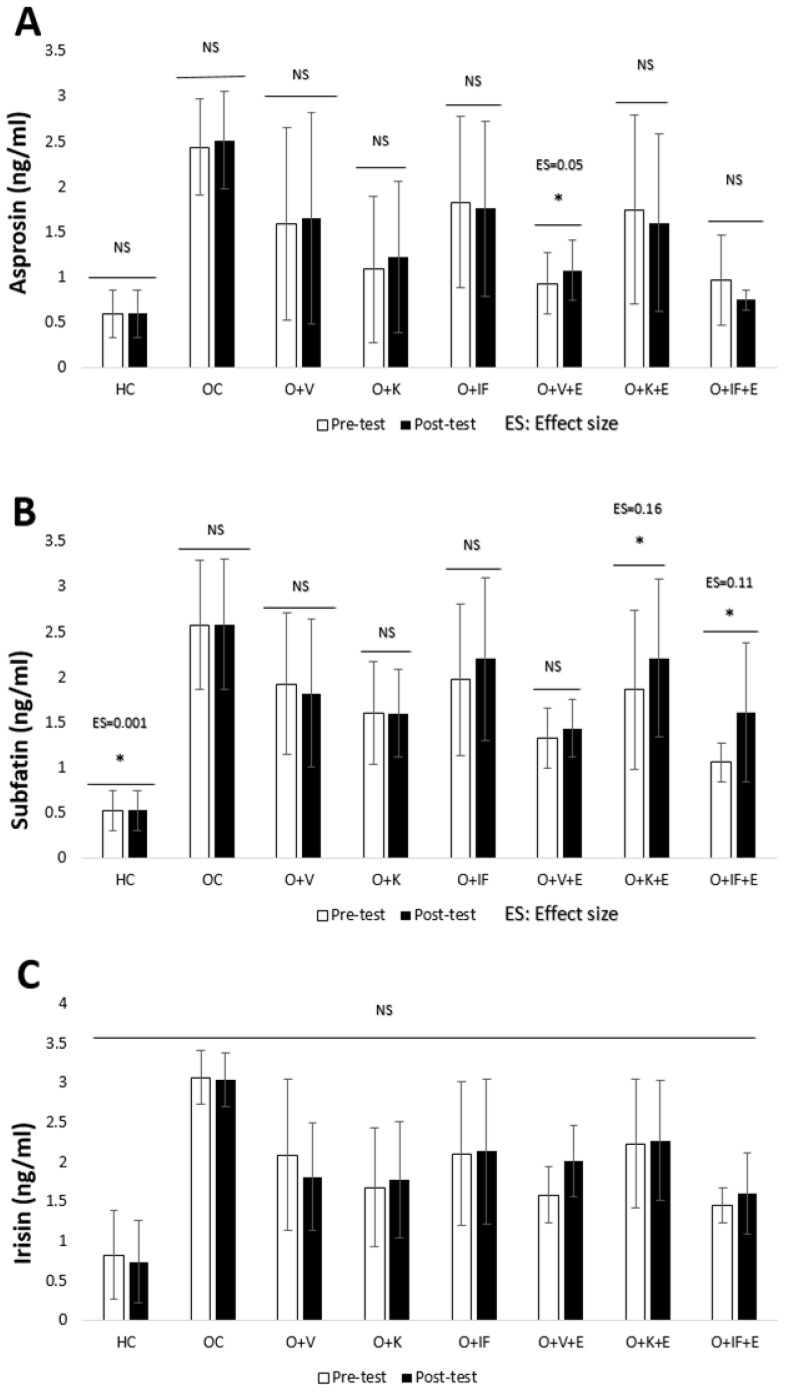
Asprosin (**A**), subfatin (**B**) and irisin (**C**) levels. Healthy Control (HC), Obese Control (OC), Obese + Vegetarian (O + V), Obese + Ketogenic (O + K), Obese + Intermittent Fasting (O + IF), Obese + Vegetarian + Exercise (O + V + E), Obese + Ketogenic + Exercise (O + K + E) and Obese + Intermittent Fasting+ Exercise (O + IF + E). *; *p* < 0.05, NS; non-significant and ES; Effect Size.

**Table 1 medicina-60-01118-t001:** Research groups.

Healthy Control (HC)	Control Group with Normal BMI
Obese Control (OC)	Control group with obese BMI
Obese + Vegetarian (O + V)	Vegetarian eating group with obese BMI
Obese + Ketogenic (O + K)	Ketogenic diet group with obese BMI
Obese + Intermittent Fasting (O + IF)	Intermittent fasting group with obese BMI
Obese + Vegetarian + Exercise (O + V + E)	Group with obese BMI, exercising and vegetarian diet
Obese + Ketogenic + Exercise (O + K + E)	Group with obese BMI, exercising and ketogenic diet
Obese + Intermittent Fasting + Exercise (O + IF + E)	Group with obese BMI, exercise and intermittent fasting

**Table 2 medicina-60-01118-t002:** Applied exercise protocols.

Movement Number	Movement Name	Number of Sets	Number of Repetitions	Minute	Explanation
1	High Knee Pulls	3	12	3–4	Arms will stand upright next to the body. The knees will be pulled to hip level in turn, the arms will swing back and forth with the movement.
2	Pulling Heels to Hips	3	20	4–5	While standing upright, one right and one left leg are pulled to the hip. It will be ensured that the heels touch the hips.
3	Leg Raises	3		4–5	While standing upright, one right and one left leg will be lifted up and tried to be brought closer to the abdominal area.
4	Squats		6	3–4	With the arms folded behind the neck, the hips will be brought closer to the ground and lifted without lifting the heels off the ground.
5	Push-Ups	2	3	4–5	The person will be laid face down parallel to the ground, and the arms will be lifted and lowered, with the toes and hands on the ground.
6	Walking Lunges	2	6 (Step)	2–3	In an upright position with hands on the shoulders, one right leg and one left leg will be pulled to the abdominal area and one will step forward.
7	Planks	2		3–4	The patient will lie face down parallel to the ground, with the toes, hands and elbows on the ground, and the right and left legs will be brought toward the abdominal area, respectively.
8	Jumping Jacks	2	12	4–6	You will jump by standing upright with your feet together and your hands at your side, put your hands together above your head and return to the starting position.
9	Sit-Ups	2	6	2–3	The participant, who is laid on her back on the ground, will pull himself toward the kneecap, without bending her knees, with his arms folded behind her neck, and return to the starting point.

## Data Availability

The data presented in this study are available on request from the corresponding authors.

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
