# Peer review of "Leptin, Ghrelin, Irisin, Asprosin and Subfatin Changes in Obese Women: Effect of Exercise and Different Nutrition Types"

_medicina, 2024, doi:10.3390/medicina60071118_

Round 1

Reviewer 1 Report

Comments and Suggestions for Authors

Obesity, being one of the most significant diseases of civilization, causes the development of a huge number of pathologies and worsens the quality and duration of life. Therefore, the search for methods of treatment of this type of lipid metabolism disorder, including pathophysiological justification, as presented in this study, makes a significant contribution to scientific and practical activities in the field of medicine. Thanks to the results obtained in this study, taking into account contraindications to a number of diets, physicians can choose a trajectory for competent reduction of excess body weight in different patients and predict the biochemical aspects of achieving favorable changes in a number of indicators.

 The material is presented well, according to accepted standards, clearly. The results obtained cannot be interpreted ambiguously. Everyone involved in the execution of this study has worked hard.

I have a few comments and questions for the authors, which will make the publication look perfect:

1. Line 59: The sentence is not clear: ‘These adipokines; They play a role in many physiological processes of the body such as nutrition…’ Please, rephrase.

2. Lines 66-67: Did you mean ‘irisin’ rather than ‘iris’? I believe this is a typo.

3. Line 81: ‘For this reason, it was hypothesized in the study that the combined application of exercise and nutrition models could affect biochemical elements that are important in the treatment of obesity.’ Your study does not examine the effect of obesity, diet and exercise on changes in any bioelements. Replace this phrase with a phrase that is appropriate to your work.

4. Line 92-94: ‘Exclusion criteria from the study; It is applied in cases where there is a problem during or after blood collection or an injury during the exercise program.’  What exactly were the exclusion criteria from the study? And the sentence is constructed incorrectly, please, correct it.

5. Table 2, last column: Your study included only women, so the pronoun should be used her, not his. In the 5th column, it is better to put the name in plural.

6. Figures 2, 3, 4, 5. Superscripts are more readable when you put abbreviations in brackets after the phrase rather than before it. For example: Healty Control (HC); Obese Control (OC); Obese + Vegeterian (O+V); and so on.

7. Figure 3: The captions in the figure legends are not readable. Change the font or font size.

8. Lines 223-224, 268: ‘Regular exercise and sports; In addition to reducing body fat levels and protecting against…’ and ‘Blüher et al. They found that there was a 12% increase in irisin levels …’. The sentence is constructed incorrectly. Please, rephrase.

9. Meteorin-Like (Subfatin) is already written on line 75. Based on this, enter, a similar term on lines 276, 277, 279, or vice versa.

10. The Conclusion does not fully summarize the results based on the aim of the study. Please, add the description of the effects of exercise program supported by different diets on biochemical values.

Best regards.

Author Response

We would like to thank you for the time allowed to this review process. As a result, we are submitting the revised version for a possible publication in this respectable Journal. Below, you can find our responses; each comment is followed by its respective reply. We made changes in the manuscript in order to address suggestions; we used yellow color for the responses to your comments and general modifications. All authors have made sufficient contributions and have approved the submitted manuscript.

Best regards,

The Authors

Legend:

R1(Reviewer 1)

A (Authors)

1) R1:

Obesity, being one of the most significant diseases of civilization, causes the development of a huge number of pathologies and worsens the quality and duration of life. Therefore, the search for methods of treatment of this type of lipid metabolism disorder, including pathophysiological justification, as presented in this study, makes a significant contribution to scientific and practical activities in the field of medicine. Thanks to the results obtained in this study, taking into account contraindications to a number of diets, physicians can choose a trajectory for competent reduction of excess body weight in different patients and predict the biochemical aspects of achieving favorable changes in a number of indicators.

The material is presented well, according to accepted standards, clearly. The results obtained cannot be interpreted ambiguously. Everyone involved in the execution of this study has worked hard.

A:

We thank the Reviewer 1 for the appreciation.

2) R1:

Line 59: The sentence is not clear: ‘These adipokines; They play a role in many physiological processes of the body such as nutrition…’ Please, rephrase.

A:

Thank you for your attention. The sentence has been revised.

New sentence: Adipokines play a role in many physiological processes of the organism, such as appetite, energy balance, insulin and glucose metabolism, lipid metabolism, and blood pressure regulation. One of these, asprosin,…………..

3) R1:

Lines 66-67: Did you mean ‘irisin’ rather than ‘iris’? I believe this is a typo.

A:

We are grateful for your attention. Yes, typo. Corrected to irisin.

4) R1:

Line 81: ‘For this reason, it was hypothesized in the study that the combined application of exercise and nutrition models could affect biochemical elements that are important in the treatment of obesity.’ Your study does not examine the effect of obesity, diet and exercise on changes in any bioelements. Replace this phrase with a phrase that is appropriate to your work.

A:

Thank you for your contribution. The sentence has been revised.

New sentence: For this reason, it was hypothesized in the study that the combined application of exercise and nutrition models on obese women may affect the biochemical elements associated with obesity.

5) R1:

Line 92-94: ‘Exclusion criteria from the study; It is applied in cases where there is a problem during or after blood collection or an injury during the exercise program.’  What exactly were the exclusion criteria from the study? And the sentence is constructed incorrectly, please, correct it.

A:

Thank you for your attention. The sentence has been revised.

New sentence: Exclusion criteria from the study; Participants' desire to leave the study, failure to comply with exercise programs, and illness or injury during this process were determined.

6) R1:

Table 2, last column: Your study included only women, so the pronoun should be used her, not his. In the 5th column, it is better to put the name in plural.

A:

Thank you. It is revised as her.

7) R1:

Figures 2, 3, 4, 5. Superscripts are more readable when you put abbreviations in brackets after the phrase rather than before it. For example: Healty Control (HC); Obese Control (OC); Obese + Vegeterian (O+V); and so on.

A:

The arrangements you specified have been made.

8) R1:

Figure 3: The captions in the figure legends are not readable. Change the font or font size.

A:

Image quality and readability have been improved.

9) R1:

Lines 223-224, 268: ‘Regular exercise and sports; In addition to reducing body fat levels and protecting against…’ and ‘Blüher et al. They found that there was a 12% increase in irisin levels …’. The sentence is constructed incorrectly. Please, rephras

A:

Thank you for your attention. The sentence has been revised.

New sentences: -In addition to reducing body fat levels and protecting against obesity-related chronic diseases, regular exercise and sports are also known to have therapeutic roles

  • Blüher et al. stated that after a one-year diet + exercise program, there was a 12% increase in irisin levels of obese young people

10) R1:

Meteorin-Like (Subfatin) is already written on line 75. Based on this, enter, a similar term on lines 276, 277, 279, or vice versa.

A:

It is edited.

11) R1:

The Conclusion does not fully summarize the results based on the aim of the study. Please, add the description of the effects of exercise program supported by different diets on biochemical values.

A:

Necessary explanations were added.

Author Response

We would like to thank you for the time allowed to this review process. As a result, we are submitting the revised version for a possible publication in this respectable Journal. Below, you can find our responses; each comment is followed by its respective reply. We made changes in the manuscript in order to address suggestions; we used yellow color for the responses to your comments and general modifications. All authors have made sufficient contributions and have approved the submitted manuscript.

Best regards,

The Authors

Legend:

R2(Reviewer 2)

A (Authors)

1) R2:

AUTHORS NEED TO DESCRIBE MORE EXTENSIVE THE MECHANISM OF ACTION, EVENTUALLY A GRAPHYCAL REPRESENTATION WITH LYMPHOCITE INHIBITION.

efects of Hidroxicloroquine during pregnancy in SLE patients, or having Anti Ro antibodies, and the efect on neonatal lupus risk.

more data about the efects on SARSCOV2 infection ( specific)

A:

Thank you, but these evaluations are not related to our study.